# Impact of High Light Intensity and Low Temperature on the Growth and Phenylpropanoid Profile of *Azolla filiculoides*

**DOI:** 10.3390/ijms24108554

**Published:** 2023-05-10

**Authors:** Sara Cannavò, Agnese Bertoldi, Maria Cristina Valeri, Francesco Damiani, Lara Reale, Federico Brilli, Francesco Paolocci

**Affiliations:** 1Department of Chemistry, Biology, and Biotechnology, University of Perugia, 06123 Perugia, Italy; sara.cannavo@studenti.unipg.it (S.C.); agnese.bertoldi@unipg.it (A.B.); mariacristina.valeri@ibbr.cnr.it (M.C.V.); 2Institute of Bioscience and Bioresources (IBBR), National Research Council of Italy (CNR), 06128 Perugia, Italy; francesco.damiani@ibbr.cnr.it (F.D.); francesco.paolocci@ibbr.cnr.it (F.P.); 3Department of Agricultural, Food and Environmental Sciences, University of Perugia, 06121 Perugia, Italy; lara.reale@unipg.it; 4Institute for Sustainable Plant Protection (IPSP), National Research Council of Italy (CNR), 50017 Sesto Fiorentino, Italy

**Keywords:** abiotic stress, photosynthesis, photoinhibition, *Azolla* spp., gene expression, flavonoids, growth indexes, MBW complex

## Abstract

Exposure to high light intensity (HL) and cold treatment (CT) induces reddish pigmentation in *Azolla filiculoides*, an aquatic fern. Nevertheless, how these conditions, alone or in combination, influence *Azolla* growth and pigment synthesis remains to be fully elucidated. Likewise, the regulatory network underpinning the accumulation of flavonoids in ferns is still unclear. Here, we grew *A. filiculoides* under HL and/or CT conditions for 20 days and evaluated the biomass doubling time, relative growth rate, photosynthetic and non-photosynthetic pigment contents, and photosynthetic efficiency by chlorophyll fluorescence measurements. Furthermore, from the *A. filiculoides* genome, we mined the homologs of *MYB*, *bHLH*, and *WDR* genes, which form the MBW flavonoid regulatory complex in higher plants, to investigate their expression by qRT-PCR. We report that *A. filiculoides* optimizes photosynthesis at lower light intensities, regardless of the temperature. In addition, we show that CT does not severely hamper *Azolla* growth, although it causes the onset of photoinhibition. Coupling CT with HL stimulates the accumulation of flavonoids, which likely prevents irreversible photoinhibition-induced damage. Although our data do not support the formation of MBW complexes, we identified candidate *MYB* and *bHLH* regulators of flavonoids. Overall, the present findings are of fundamental and pragmatic relevance to *Azolla*’s biology.

## 1. Introduction

*Azolla* is a small genus of aquatic ferns growing in tropical, subtropical, and temperate freshwater ecosystems. All *Azolla* species live in symbiosis with microorganisms, among which the nitrogen-fixing cyanobacterium *Trichormus azollae* is the most relevant [1]. Through the action of nitrogenase enzymes, *T. azollae* reduces the atmospheric nitrogen to ammonium, which is taken up by the fern and, upon decomposition, is released into the soil, where it is converted into nitrate and nitrite [2]. This is why *Azolla* spp. have been used as biofertilizers in rice paddies in southeast Asia for several centuries [3].

*Azolla* spp. can reach high growth rates and, depending on the growing conditions, can double their biomass in 3–10 days [4]. Thus, the relative growth rate (RGR) of *Azolla* is much higher than that of other land plant species [5,6]. Studies on the effects of single environmental variables on *Azolla* RGR started in the 1960s [7] and mainly focused on the effects of temperature [8]. The sustained growth of *Azolla* is supported by high CO_2_ photosynthetic assimilation rates [9]. Although *Azolla* requires low light to saturate the electron transport rate (ETR) and optimize photosynthesis [10,11], it can acclimate to higher light intensities [12]. This effect occurs through an efficient non-photochemical quenching process, which dissipates excess light energy in the form of heat [13]. Besides the temperature and light intensity, the medium composition and cultivation system play crucial roles in RGRs. As in many other photosynthetic aquatic organisms, phosphorus is the primary nutrient limiting the growth of *Azolla* spp. [14]. The RGR of *Azolla filiculoides* also depends on nitrogen availability, which might be different depending on whether these ferns are cultivated in continuous or batch production systems [15,16,17,18]. Changes in *Azolla* growth rate as well as in the content of chlorophylls, carotenoids, and phenolic compounds have also been investigated in response to the presence of pollutants, such as copper, arsenic, and cadmium [19,20,21].

Thanks to the elevated growth performance and the high levels of proteins and lipids that can be synthesized in their small leaves, which are defined as fronds, *Azolla* spp. are also regarded as feed and biofuel crops [22]. Moreover, the amino acid profile of *Azolla* proteins is similar to that of soybean [23], the crop that most largely contributes to livestock feed. However, the high levels of feed-deterrent flavonoids contained in *Azolla* fronds have discouraged the use of this species as a sustainable source of proteins for animal consumption. In fact, the digestibility and bioassimilability of *Azolla* forage are seriously compromised by the presence of secondary metabolites, such as condensed tannins [24]. Along the same vein, we recently showed that the concomitant decrease in temperature and increase in light intensity triggers severe remodeling in the metabolome of *A. filiculoides* fronds, particularly enhancing red pigments such as phlobaphenes and 3-deoxyanthocyanidins [25]. Keeping *A. filiculoides* for 20 days under a lower temperature and higher light intensity induced differential regulation between early and late gene families of the phenylpropanoid pathway. The latter (i.e., *dihydroflavonol reductase 1* and *2*, *DFR1* and *DFR2*) were already overexpressed after 3 days of treatment [25].

In angiosperms, as in liverworts and mosses, individual application of low temperature and high light intensity induce reddening, although this effect is much stronger when these two conditions are combined [26,27,28,29,30,31,32]. However, how the application of either high light intensity or low temperature and their combination impact photosynthetic performances, growth rate, and pigment accumulation in *A. filiculoides* cultures still needs to be better elucidated in order to fully exploit the potentiality of this species as fodder, biofuel, or nutraceutical crop [15].

Dissecting the contribution of environmental conditions to the biosynthesis of pigments in *Azolla* is also of fundamental relevance. In higher plants, the expression of the late biosynthetic genes (LBGs) of the phenylpropanoid pathway, particularly those involved in the accumulation of flavonoid pigments, such as anthocyanins and proanthocyanidins (PAs), is dependent on several environmental factors [33]. The ternary transcriptional regulatory complex MBW, which is composed of MYB, bHLH, and WD40 proteins, mediates the expression of LBGs according to the external conditions [34,35,36,37]. Conversely, flavones, flavonols, and phlobaphenes are regulated by single MYBs [38,39]. Our understanding of the genetic regulation of the flavonoid pathway in lower plant species is still poor [40]. There is evidence showing that in liverworts, the *R2R3MYB* and *bHLH* genes regulate flavonoid biosynthesis, while *WDR* genes are present in their genome [41]. However, whether the expression of LBGs in bryophytes and other lower vascular plant species (i.e., ferns) is controlled by the MBW complex, and which are the most critical regulatory proteins, are questions that remain to be addressed. The release of the *A. filiculoides* genome has propelled whole transcriptome analyses on genes related to nitrogen metabolism [15,42] and allowed the first evidence on the organization, evolution, and function of key genes related to the phenylpropanoid pathway to be gained (Figure 1) [25,43,44].

These genetic tools, coupled with the metabolic plasticity of this species, render *A. filiculoides* a model to address the genetic x environmental interactions underlying the biosynthesis of a vast array of secondary metabolites.

Here, to disentangle the impact of temperature, light intensity, and their combination on the growth rate, photosynthetic performance, and profiles of the main classes of phenylpropanoids, *A. filiculoides* was grown in batch culture and exposed to higher/lower temperatures and higher/lower light intensities. Moreover, candidate *MYB*, *bHLH*, and *WDR* regulatory genes were mined from the *A. filiculoides* genome, and their expression levels, along with those of key LBGs, were monitored under these different temperature and light conditions. Thus, this study provides novel information on how higher/lower light intensities and temperatures affect the growth of *A. filiculoides* as well as investigating the genetic and environmental determinants underlying the accumulation of flavonoid pigments in this plant. The presented findings are crucial to exploit *A. filiculoides* as a potentially innovative crop and model fern species to unearth the regulation of secondary metabolites.

## 2. Results

### 2.1. Effects of Different Light and Temperature Conditions on the Growth of A. filiculoides

*Azolla* plants grown for 20 days under control conditions (CNT) increased their biomass by more than 8.6-fold (Figure 2A). When plants were challenged with high light intensity (HL) treatment, the increment in biomass production was even higher (34-fold), while an increase, although not significant, was observed in plants that experienced the cold treatment (CT), regardless of the light intensity (Figure 2A). A 7-day recovery period further enhanced the biomass of *Azolla*. However, the relative increment of biomass was similarly high in plants following either HL or CNT conditions, and it was equally low following both CT and high light intensity and cold treatment (HLCT) (Figure 2A).

After 20 days, *A. filiculoides* displayed a lower doubling time (Dt) value in HL (3.94 ± 0.04) than in CNT conditions (6.45 ± 0.18). Conversely, this value increased significantly in the CT (174.25 ± 3.61) and HLCT (69.75 ± 8.15) groups (Figure 2B). Following recovery, the Dt lowered in all the treated plants. However, it remained higher in *Azolla* plants that underwent CT (11.48 ± 0.30) and HLCT (11.59 ± 0.65) than in plants grown under CNT conditions (Figure 2B).

The RGR significantly increased only in *Azolla* following HL treatment, whereas it declined in plants grown under CT, regardless of the light intensity, compared to the CNT group (Figure 2C). Following recovery, the RGR increased in any treated *A. filiculoides*, although it was the lowest in plants treated with CT, and the highest in those treated with HL (Figure 2C).

In addition, the dry weight (DW) was significantly higher in plants that experienced HL and CT. The HLCT increased the DW even further, which resulted in a value more than 2-fold higher than that in CNT plants (Figure 2D).

### 2.2. Photosynthetic Activity and Accumulation of Chlorophylls and Carotenoids under Different Light and Temperature Conditions

Exposure of *Azolla* to HL did not affect the chlorophyll fluorescence light response curves of the ETR (Figure 3A). After CT, the values of both the ETR and the maximum PSII quantum efficiency (ΦPSII) were reduced in parallel with those of the fraction of open PSII centers (qL) (Figure 3A–D) and the maximum Fv/Fm (Table 1). All of these parameters dramatically decreased under HLCT. However, while the application of CT did not alter the rate constant for the heat dissipation of chlorophyll excitation energy (non-photochemical quenching, NPQ), the coupling of CT with HL decreased the light response of NPQ to a constant lower value when measured at light intensities > 200 µmol^−2^ s^−1^ (Figure 3B).

The application of CT did not significantly influence the contents of chlorophyll a (Chl a) or chlorophyll b (Chl b) with respect to *Azolla* plants grown under CNT; as a consequence, the Chl a/Chl b ratio did not change (Table 2). On the contrary, the levels of both Chl a and Chl b decreased significantly after either HL or HLCT, and the decrease of Chl b was more marked in HL-treated plants. Overall, the Chl a/Chl b ratio was lower in plants that experienced HLCT, which also showed the lowest levels of carotenoids (Table 2).

### 2.3. Impacts of Light Intensity and Temperature on the Polyphenolic Profile of A. filiculoides

The pigmentation of *Azolla* fronds changed qualitatively after experiencing the 4 different treatments: the fronds remained bright green under CNT (Figure 4A) as well as after CT (Figure 4B), while they turned light green with reddish edges after HL (Figure 4C) and became almost entirely reddish after HLCT (Figure 4D).

The results of spectrophotometric assays showed that HL conditions caused significant increases in anthocyanidins, apigeninidins, luteolinidin 5-O glucoside, and soluble proanthocyanidins, regardless of the temperature (Table 3). The application of CT did not significantly change the polyphenolic levels, except for insoluble proanthocyanidins, which decreased in concentration with respect to CNT plants (Table 3). In stark contrast, the combined application of cold and high light intensity in HLCT triggered a significant increase in the content of phlobaphenes. These pigments almost doubled in content and had concentrations 1.4-fold higher in plants after HLCT than in those exposed to CNT and HL treatments. Differently, HLCT did not stimulate any significant further increases in the other polyphenols with respect to HL treatment, while it induced a significant decrease in phenolic acids (Table 3).

### 2.4. Effects of Light and Temperature on the Expression of Key LBGs

The key LBGs, *DFR* and *LAR*, have been already annotated and/or functionally characterized in *Azolla* ([43,44], Figure 1). Thus, we investigated whether the expression of these genes changed following the treatments applied to *Azolla*. The members of the *DFR* gene families *DFR1* and *DFR2* differed in their expression extents and patterns (Figure 5A). The two genes of *DFR1* family showed higher mRNA levels than those in *DFR2*, regardless of the treatment. The mRNA levels of *DFR2-2* remained stable among the treatment groups, while those of *DFR2-1* were the highest after the application of CNT and HL and the lowest after CT. Conversely, *DFR1-1* responded positively to the HL treatment only. At the same time, *DFR1-2* was significantly and highly upregulated under HLCT, whereas it showed minimal levels under low light conditions, regardless of the temperature. *LAR* was similarly upregulated in plants after applying HL and CT, and its expression decreased in HLCT to a level slightly higher than that in CNT (Figure 5A).

### 2.5. Genome Mining and Expression Analysis of the Flavonoid Pathway Regulators in A. filiculoides

In *A. filiculoides*, *R2R3MYBs* of the *MYB III D* and *E* classes that regulate the expression of phenylpropanoid genes and cell patterning processes, respectively, in higher plant species have been already annotated and investigated [43]. However, to our knowledge, neither *WDR* nor *bHLH* members have been studied thus far in *A. filiculoides*. To fill this gap, we analyzed the *A. filiculoides* genome for the presence of putative homologs of *WDRs* and *bHLHs* shown to be involved in the control of the flavonoid pathway in other plant species [35]. As for the WDR component, the BLASTP search with TTG1s yielded at least 6 proteins showing the characteristic WD-40 repeat domains. The same proteins were employed to build a phylogenetic tree along with WDR proteins from higher plants, to which the *Azolla* WDR showed the highest sequence similarity. Two *Azolla* proteins, namely Azfi_s0016.g014399 and Azfi_s4235.g118410, formed a sister subclade to the one containing the LWD1 protein from *Cicer arietinum* and the two TTG1s TTG1-1 and TTG1-2, which regulate the phenylpropanoid pathway in *Rubus genevieri* ([45]; Appendix A). Conversely, the remaining four *Azolla* WD-40 repeats containing proteins, in keeping with their original annotation as transducing-like, clustered with the *Arabidopsis thaliana* WDR55 protein (NP 973596.1) belonging to the transducing family (Appendix A). Based on these findings, only the *Azolla* Azfi_s4235.g118410 and Azfi_s0016.g014399 proteins were further considered in our study and were named TTG1-1 and TTG1-2, respectively.

The top 6 *Azolla* proteins retrieved from the BLASTP search for homologs of bHLH flavonoid regulators from either higher (i.e., the *Arabidopsis* TTG8, EGL3, GL3 [46,47]) or lower (i.e., the *Plagiochasma appendiculatum* bHLH1, [48]) plants displayed only a limited identity to the queries.The phylogenetic tree built from the alignment of several bHLHs from different plant species confirmed the evidence shown above: none of the *Azolla* bHLHs retrieved from the BLAST analysis clustered with the flavonoid regulatory bHLHs used as queries. Rather, the *Azolla* bHLHs were grouped with those involved in hormonal signaling (Appendix A).

When the two *WDR* genes were assayed, it emerged that *TTG1-1* was more highly expressed than *TTG1-2* in all treated *Azolla* plants. Moreover, *TTG1-1* was upregulated after the application of CT, whereas the expression of *TTG1-2* peaked after HL (Figure 5B). Only the *Azfi_s0018.g014776* gene, among the 6 *Azolla bHLHs* assayed, did not show a similar expression extent across the treatment groups: the mRNA steady-state levels of this gene dropped under HL conditions when only phlobaphenes decreased significantly (Figure 5B and Appendix A). Out of the 6 selected *R2R3MYBs* tested, only two showed differential expression across the treatment groups, namely *Azfi_s0001.g00083* and *Azfi_s0129.g04885* (Figure 5B and Appendix A). *Azfi_s0001.g00083* displayed the highest expression levels among the other *MYBs* under all treatments. Moreover, the mRNA levels of this regulator were the highest after HLCT and the lowest after CNT. Conversely, *Azfi_s0129.g04885* was stimulated by low temperatures, since its expression was significantly higher after CT, irrespective of the light intensity. The correlation analysis between the expression of candidate *MYB* regulators and those of the structural genes investigated showed that *MYB Azfi_s0001.g00083* is positively correlated with *DFR1-2*, but negatively correlated with *DFR2-1*, while *Azfi_s0129.g04885* is negatively correlated with both *DFR1-1* and *DFR2-1*. Among the *bHLH* and the *WD-40* genes, the expression of *bHLH Azfi_s0018.g014776* is negatively correlated with both *DFR1-1* and *DFR2-2*, while that of *TTG1-2* is positively correlated with *DFR1-1* (Table 4).

## 3. Discussion

### 3.1. Light and Temperature Impact the Photosynthetic Activity and Growth of A. filiculoides Differently

Our photochemistry measurements confirm that, in *A. filiculoides*, photosynthesis is optimized under low light conditions [10,11] as, consistent with the results of Shi and Hall [49], the light-curve response of the ETR saturates at an intensity of only 400 μmol m^−2^ s^−1^, regardless of the growing temperature. Moreover, the same analyses confirmed that *Azolla* can successfully adapt to an increasing light intensity [12,50], although this adaptation highly depends on the temperature. In our experiments, exposure to a higher light intensity (700 μmol m^−2^ s^−1^) reduced the content of chlorophyll in *A. filiculoides* grown at 25 °C, although this did not affect the photochemical processes of photosynthesis assessed through fluorescence measurements (under both light and dark conditions). Indeed, the same light and temperature conditions induced faster growth and higher biomass production. In particular, exposure to higher light intensities had a long-lasting impact on the development of *A. filiculoides*, which continued when the light intensity was reduced during the recovery period. A higher light intensity could prolong the stimulation of the biomass production by fulfilling the energy requirements to maximize the biochemical processes of photosynthetic CO_2_ assimilation (i.e., the Calvin Benson Cycle, Ribulose-1,5-bisphosphate carboxylase/oxygenase activity), which in *Azolla* spp., are as high as in C4 plant species [51]. It is also conceivable that high light exposition for 20 days could have been sufficient to modulate the phytohormone signaling pathways [52], which further enhanced growth during the recovery phase. Consistent with this observation, following the recovery phase, the RGR was still higher in plants exposed to an increasing light intensity, although the Dt values remained similar. At a higher light intensity and 25 °C, the growth of *A. filiculoides* could have been constrained by the nitrogen availability, since the nitrogen reductase activity of the N_2_-fixing cyanobacterium *T. azollae* has been demonstrated to be saturated under low light conditions [53]. Moreover, self-crowding, which might have occurred in our production system, could have limited the growth rate potential of *A. filiculoides*, which did not double the amount of biomass in the length of time reported in the literature [16,54]. Nevertheless, we cannot exclude the possibility that the temporal window of our biometric surveys (20 days) was not long enough to capture significant variations of the biomass doubling time under the growth conditions employed. Notwithstanding, the low light intensity and chlorophyll requirements for the optimal functioning of photosynthesis, together with the capacity to acclimate to a wide range of light intensities, confirm the ability of *A. filiculoides* to quickly colonize various environments [55,56].

Our results also highlight that the growth of *A. filiculoides* is impaired by cold treatment (5 °C). Photosynthesis and growth are downregulated by low temperatures [57]. Previous studies have reported that the optimal growth temperature ranges from 25 to 30 °C for *A. filiculoides* under low light intensities [58,59]. Here, we showed that low temperatures reduce the photosynthetic efficiency, as indicated by decreased slopes in the initial parts of both the ETR and PSII light response curves. This drop was exacerbated when cold treatment was applied under a higher light intensity, as is discussed in the next paragraph. However, under low temperature conditions, the light-curve response of the ETR was always saturated at ~400 μmol m^−2^ s^−1^, and only a moderate decrease in the maximal PSII quantum yield (Fv/Fm) was measured after dark adaptation. On the other hand, under a lower light intensity, the NPQ and qL fluorescence parameters showed similar responses in *A. filiculoides* exposed to either 25 °C or 4 °C, highlighting that temperature only slightly affects the non-photochemical quenching process. The cold treatment applied in our experiments might have induced the onset of photoinhibition, as the accumulation of fresh biomass was hampered. Nevertheless, the low temperature, per se, slightly influenced PSII (i.e., decreasing the maximum quantum efficiency by ~18%), and consistently with what has been reported by Janes [60], it did not permanently affect *Azolla* growth. In fact, after recovery from the cold treatment, the Dt values decreased and those of RGR increased, regardless of the light conditions.

Finally, it is worth noting that the dry weight of *Azolla* peaked under combined conditions of low temperature and high light. This increase in dry matter could have resulted from the accumulation of cell-wall-bound components, such as phlobaphenes (Table 3), and/or carbohydrates. However, it is known that cold can increase the cell wall thickness as well as the content of carbohydrates [61,62]. Thus, the increase in dry weight observed in *Azolla* under the combination of a low temperature and high light intensity might have resulted from the downregulation of both the photosynthetic process and growth [62] due to the alteration in the carbohydrate balance, with these compounds being more rapidly produced and then translocated from *Azolla* fronds.

### 3.2. Uncoupling the Effects of Light and Temperature on the Accumulation of Phenylpropanoids

*Azolla* is able to accumulate a vast array of secondary metabolites, many of them relevant for human health [24]. In particular, the environmental conditions of light and temperature shape *Azolla*’s phenylpropanoids profile [25]. Phenylpropanoids are key indicators and mediators of plant responses to a wide range of biotic and abiotic stress stimuli [63]. As an example, UV-absorbing flavonoids located in epidermal cells strongly reduce highly energetic solar wavelengths that generate reactive oxygen species (ROS), thereby avoiding the occurrence of photooxidative stress and damage [64] Additionally, nuclear-located flavonols might chelate Fe and Cu ions, thus avoiding the generation of highly reactive hydroxyl radicals, as H_2_O_2_ may freely diffuse out of the chloroplast under severe light stress conditions [65]. Collectively, flavonoids, as well as phenolic acids, represent a class of bioactive metabolites that might even outperform the enzymatic antioxidant defense system [65,66,67,68,69,70]. Thus, the significant increases in the levels of anthocyanins and 3-deoxyanthocyanins in *A. filiculoides* challenged by a high light intensity could be explained by their light screening properties [71]. The overproduction of ROS, which affect many cellular functions, is also induced by low temperatures, as these are harsh environmental conditions that elicit the accumulation of phenylpropanoids in plants [72]. Indeed, the sole lowering of temperature was insufficient to trigger significant changes in the levels of the phenylpropanoids tested in this study with the exception of insoluble PAs. However, plant acclimation to cold is the result of complex cross-talk between a low temperature and light intensity [73], and we note that coupling a low temperature with an increasing light intensity induced a remarkable accumulation of phlobaphenes only. The current study shows that more than the other classes of pigments reported thus far [74,75], phlobaphenes are the main component responsible for the pronounced leaf reddening of *Azolla* fronds that is visible upon treatment with a high light intensity and low temperature. The increase in phlobaphenes under the conditions above leads us to argue that these compounds not only act as potential plant protectants against excessive light radiation [76,77,78] but might also contribute, in virtue of their antioxidant capacity [79,80,81,82], to the scavenging of excess ROS. Finally, it is also interesting to highlight the opposing responses between phobaphenes and phenolic acids: the latter compounds decreased in concentration under a low temperature and high light intensity, suggesting that these conditions cause severe reprogramming of the phenylpropanoid pathway in *Azolla* and that phenolic acids are not likely to play a relevant defensive role.

On the one hand, the higher content of polyphenolic compounds (i.e., phobaphenes) induced by a high light intensity at a low temperature might have avoided irreversible photooxidative damage to the reaction centers of PSII due to enhanced ROS production stimulated by cold treatment near PSII. On the other hand, these pigments might have interfered with the measurement of chlorophyll fluorescence parameters [83]. Therefore, the drop in all photosynthetic parameters monitored under low temperature and high light intensity treatment conditions might not reflect the severe and irreversible photoinhibition. This hypothesis is confirmed by the observation that, after 7 days of recovery, the biomass and growth parameters were similar to those when *A. filiculoides* was exposed to 20 days of either low temperature/low light intensity, or low temperature/high light intensity conditions. Additionally, the absence of photoinhibition-induced damage is corroborated by the evidence that the chlorophyll contents, in *Azolla* grown under low temperature and high light intensity conditions, were not different than those in *Azolla* plants treated under high temperature conditions. The latter is the condition that maximizes *Azolla*’s growth.

### 3.3. A. filiculoides: A Model Organism to Unveil the Regulation of Phenylpropanoid Biosynthesis in Lower Plants

The families of *DFR* genes likely play a central role in controlling the metabolic flux of the phenylpropanoid pathway in *Azolla*. These genes not only code for enzymes that reduce dihydroflavonols to leucoanthocyanidins, the substrates for the biosynthesis of anthocyanins and PAs, they also code for enzymes that reduce flavanones. DFR enzymes of certain plant species possess, in fact, additional flavanone 4-reductase (FNR) activity, allowing them to give off flavan 4-ols [84,85]. The latter, in turn, are the substrates for the accumulation of 3-deoxyanthocyanidins and phlobaphenes via enzymatic reactions that have not been fully elucidated yet (Figure 1). The presence of two distinct gene families, *DFR1* and *DFR2,* in the *A. filiculoides* genome [44] points towards their functional diversification. In a previous study, we showed that the two *DFR1* genes, as well as the two *DFR2* ones assayed collectively, were upregulated by cold treatment and a high light intensity [25]. Here, by disentangling the contribution of each gene member within the two families and by challenging *Azolla* with different growing conditions, we found the following: (a) both genes of the *DFR1* family were more highly expressed than those of *DFR2* and (b) each gene member of the two families behaved differently. Notably, while the expression pattern of *DFR1-1* gene mirrored and could explain the light-induced accumulation of anthocyanins and deoxyanthocyanins, that of *DFR1-2* could explain the accumulation of phlobaphenes. Indeed, the latter gene peaked under high light intensity and low temperature conditions, maximizing the accumulation of these compounds. Differently, cold treatment negatively regulated the expression of *DFR2-1*. The presence of various *DFR* members and their differential regulation according to the light and temperature environmental conditions owe have shown in *A. filiculoides* are consistent with what has been found in higher plant species, in which *DFR* gene members might differentiate from each other by their substrate-specificity and environmental and genetic regulation [86,87]. Thus, the present study paves the way for future analyses to investigate the possible functional diversification among the different *Azolla DFR* genes. Conversely, Gungor and colleagues [43] have already conducted a functional analysis of the *Azolla* LAR protein, showing its capacity to synthesize catechin, the flavan-3-ol which, together with epicatechin, is one of the building blocks of PAs. However, an additional function of LAR, at least in higher plant species, is that of limiting the length of PA polymers, thereby controlling the accumulation of insoluble PAs [88]. Consistent with this possible role, the present study showed that *LAR* expression was the lowest under the conditions of a lower light intensity and higher temperature when the level of the insoluble fraction of PAs peaked and the soluble fractions of these pigments were minimal.

In higher plant species, the expression pattern of the LBGs of the phenylpropanoid pathway, such as *DFR* and *LAR*, is controlled by the ternary transcriptional MBW complexes. These complexes are also known to regulate the development of trichomes and root hairs as well as the accumulation of seed mucilage in Arabidopsis [89]. Within this complex, the MYB partner confers specificity towards the target pathway. Conversely, the bHLH and WDR components are less specific [47]. The reduced expression of the *bHLH* and *MYB* genes likely limits the range of their regulatory activities concerning angiosperms and gymnospers [90]. Additionally, whether these regulators act singularly or through the formation of complexes to control the accumulation of flavonoids remains unclear. Nevertheless, data from *Marchantia* suggest that the MYB activators of phenylpropanoid metabolism act outside the MBW complex [40]. On the other hand, *Marchantia* accumulates flavones, and its red pigmentation results from an increasing content of the cell-wall-localized anthocyanidin aglycone riccionidin A. Nevertheless, *Marchantia* lacks the vast array of pigments described not only in angiosperms but also in *Azolla* spp. [25]. Thus, our plants, which had different pigmentation patterns under changing light and temperature conditions, allowed us to sort among the candidate regulators of flavonoids retrieved from the *A. filiculoides* genome. Concerning the bHLHs, our phylogenetic analysis does not support the presence of *Azolla* orthologs in the bHLH flavonoid regulators used as queries. However, it shows that the *Azolla* bHLHs are closely related to bHLH13s from *Malus domestica* and eggplant and to the MYC2 and MYC3 from *Arabidopsis*. The eggplant bHLH13 regulates the anthocyanin pathway by binding to the promoter of the EBGs *CHS* and *F3H*, as well as flowering by binding to *FT*, a key gene in this process [91]. Moreover, once overexpressed in *Arabidopsis*, *bHLH13* from eggplant reduces JA-induced anthocyanin accumulation [92]. Likewise, MYC2 and MYC3 regulate different subsets of the jasmonic acid (JA)-dependent transcription response [93], and JA induces the degradation of JAZ proteins with the subsequent releases of bHLH and MYB transcription factors, which promote anthocyanin accumulation [94]. Therefore, it is conceivable that the *Azolla* bHLHs investigated herein participate in the control of the flavonoid pathway. These might do so by acting on early genes of the pathway and/or under stress signals transduced by phytohormones. Along this line of reasoning, we note that *Azfi_s0018.g014776* is downregulated under higher light intensity conditions to the same extent as the *DFR1-1* and *DFR2-2* genes. Whether the downregulation of this *bHLH* and that of the two structural genes is related to different hormonal levels in *Azolla* will be a matter of future investigations.

Overall, our gene expression analyses do not point to the involvement of the selected members of the *bHLH* and *WDR* families in the regulation of *Azolla* pigmentation via the MBW complex. This is also consistent with the fact that none of the *Azolla* MYB proteins investigated retain the conserved amino acid signature ([D/E]LX2[R/K]X3LX6Lx3R) involved in the interaction with the bHLH to form the MBW complexes [95]. In keeping with data from Güngör et al. [43], who investigated class IIID and class *IIIE* MYBs during the diel cycle, we report that the two *IIID* genes were much more highly expressed than the four III E, regardless of the treatment. Moreover, here we show that *Azfi_s0001.g00083* was upregulated under lower temperature and higher light intensity conditions, especially when these two conditions were applied simultaneously, and its expression correlated with that of *DFR1-2*, which might represent its target among the *DFR* gene families. Cold treatment coupled with a high light intensity maximizes the accumulation of phlobaphenes and reduces, at minimum, that of phenolic acids. Therefore, *Azfi_s0001.g00083* could be regarded as the putative cold- and light-induced MYB regulator responsible for the flux diversion of the phenylpropanoid pathway towards phlobaphenes at the expense of upstream products, such as phenolic acids. After embracing this hypothesis, one could argue that, in *Azolla*, the genetic control of phlobaphenes could depend on a single *MYB* only, as occurs in maize [96]. We cannot exclude the idea that the other *MYBs* here considered could be modulated according to different light and temperature conditions and/or at earlier time points than those investigated here. Notwithstanding, our analyses suggest that *Azfi_s0001.g00083* and the cold-induced *Azfi_s0129.g04885* might participate in the regulation of flavonoids. The dynamics of the expression patterns of these two *MYBs* and *bHLH Azfi_s0018.g014776* at earlier time points along with their ectopic expression in higher plant models will help us to disclose more about their functions. Likewise, the ongoing analysis of the whole transcriptomes of *Azolla* plants under different light and temperature conditions will permit us to unveil more about the structural and regulatory genes underlying the biosynthesis of flavonoids.

In conclusion, our findings show that temperature has a more significant impact than light intensity on the constraint of *A. filiculoides* growth and show how light plays a key role in enhancing the production of flavonoids, particularly those involved in reddish pigmentation, as a mechanism to prevent irreversible damage to PSII induced by photoinhibition when a higher light intensity is coupled with cold treatment. Moreover, we gained a first glimpse of the genetic regulators of pigments and their interactions with environmental factors. Our observations lay the groundwork for future investigations to optimize the environmental conditions, allowing the exploitation of *Azolla* as a sustainable biofactory of secondary metabolites. This information is relevant to human and animal health and could be used to better understand the regulation of the phenylpropanoid pathway in this plant species.

## 4. Materials and Methods

### 4.1. Experimental Plant Material

The specimens of *Azolla* employed in this study were collected in July 2020 from a small pond in the botanical garden of the University of Perugia, Italy (latitude 43°05′52″ N, longitude 12°23′49″ E) and further identified as *A. filiculoides* (Lam.) using molecular markers [25]. Once transferred in the laboratory, the plants were grown in batch culture in a climatic chamber under the following conditions: 10 h photoperiod, 25/20 °C day/night temperatures, and a photosynthetic photon flux density (PPFD) of 220 μmol m^−2^ s^−1^ provided by fluorescent tubes. The growing medium was the nitrogen-free nutrient reported by Watanabe and colleagues [97], which was replaced once per week to avoid nutrient limitations.

### 4.2. Experimental Treatments and Sampling for Metabolic and Molecular Analyses

Once acclimated as reported above, four *A. filiculoides* pools were collected from three independent batch cultures and exposed to the following four conditions for 20 days: (1) 14 h photoperiod, 25/20 °C day/night temperatures, and PPFD = 220 μmol m^−2^ s^−1^ as the control (CNT); (2) 14 h photoperiod, 25/20 °C day/night temperatures, and PPFD = 700 μmol m^−2^ s^−1^ as the high light intensity (HL) treatment; (3) 14 h photoperiod, constant 5 °C temperature, and PPFD = 220 μmol m^−2^ s^−1^ as the cold treatment (CT); (4) 14 h photoperiod, constant 5 °C temperature, and PPFD = 700 μmol m^−2^ s^−1^ as the combined high light intensity and cold treatment (HLCT). Following the treatments, all plants were grown under CNT conditions for 7 days during the recovery period. The entire experimental set-up was replicated twice with three replicates per treatment each time. The design was completely randomized (CRD). At the two sampling times, after 20 days of treatment and after 7 days of recovery, *A. filiculoides* plants were rinsed with sterile water, collected, and directly employed for the analysis of growth parameters and the evaluation of the contents of chlorophylls and carotenoids. Conversely, for the phenylpropanoid and gene expression analyses, fronds were detached from the roots of rinsed plants with a blade, weighed, frozen in liquid N_2_, and either freeze-dried for the analysis of the phenylpropanoids or stored at −80 °C for RNA extraction.

### 4.3. Evaluation of Growth Parameters

In order to evaluate the growth parameters under different treatments, ~3 g of *A. filiculoides* under CNT conditions was employed for each of the 3 replicates per treatment. After 20 days of treatment, *A. filiculoides* plants were gently dried with paper and weighed to assess their biomass increase, Dt, and RGR. Once weighed, the plants were grown under CNT conditions for seven days during the recovery period. Then, the plants were collected, gently dried with paper, and the fresh weights were measured once again. The Dt and RGR parameters were calculated using the equations described in [98,99], as follows:Dt = 0.693 t/ln (B_f_/B_0_)
where B_f_ is the final biomass, B_0_ is the initial biomass, and t is the growth period.
RGR = (lnB_2_ − lnB_1_)/(t_2_ − t_1_)

B_1_ represents the plant biomass at time t_1_ (0 or 20 days) and B_2_ is the biomass at time t_2_ (either 20 or 27 days). Dt and RGR were expressed in days (d) and g/g d, respectively.

The dry weight achieved after each of the 20-day treatments was assessed by sampling ~150 mg of *Azolla* plants from each replicate. Then, they were desiccated at 90 °C in an oven until they reached a constant weight.

### 4.4. Analysis of Chlorophylls, Carotenoids, and Phenolic Compounds

The contents of chlorophylls (Chl) and carotenoids (Car) were determined by the sampling of 100 mg of fresh material, which was mechanically broken with quartz until a fine powder was obtained. Then, samples were extracted twice with 80% (*v*/*v*) acetone and centrifuged at 3000 rpm (4 °C) for 15 min. The supernatants were pooled and read at the spectrophotometer at 661 nm for Chl a, 644 nm for Chl b, and 470 nm for Car. The concentrations of these pigments (μg g^−1^ fresh weight) were calculated using the following equations [100]:[Chl a] = [12.21 × (E661 − E750)] − [2.81 × (E644 − E750)]
[Chl b] = [20.13 × (E644 − E750)] − [5.03 × (E661 − E750)]
[Car] = [(1000 × (E470-E750)] − (3.27 × [Chl a]) – (104 × [Chl b])/198

In order to exclude the contribution of the residual scattering of the acetone solution, the absorbance (E) for each sample and pigment were corrected by subtracting those of the buffer read at 750 nm.

With minor modifications, anthocyanidins and deoxyanthocyanidins were extracted from *Azolla* plants as described by Dong et al. [101] and Cohen et al. [102]. Aliquots of freeze-dried *A. filiculoides* fronds ranging from 10 to 15 mg were ground in liquid nitrogen and extracted overnight with 1.5 mL of 1% (*v*/*v*) HCl in methanol at 4 °C and under continuous agitation. These extracts were then centrifuged at 13,000 rpm and 4 °C for 10 min, and 1 mL of distilled water added to the supernatant. Anthocyanidins were read with the spectrophotometer at 530 nm and quantified as cyanidin 3-glucoside equivalents (ɛ = 26,900 L m^−1^ mol^−1^, MW = 484.82). Moreover, the extracts obtained as above were read at 479 nm for apigeninidin (ɛ = 38,000 L m^−1^ mol^−1^, MW = 255.24) and 496 nm for luteolinidin-5-O-glucoside quantification, as reported in Cohen et al. [102].

Spectrophotometric determination of phenolic acids and flavonols was carried out in accordance with Cassani et al. [78] with some modifications. Samples of ~15 mg of freeze- dried *Azolla* fronds were first boiled with 200 μL of distilled water for 30 min and then left at 4 °C under agitation overnight with 1.5 mL of solution (1% HCl, 95% ethanol). These extracts were centrifuged at 13,000 rpm and 4 °C for 10 min, and both flavonols and phenolic acids were quantified from the supernatant through the absorbance at 350 nm and 280 nm, respectively. In particular, the concentration of flavonols was calculated as quercetin 3-glucoside equivalents (ε = 21,877 L m^−1^ mol^−1^, MW = 464.82), and that of phenolics was calculated as ferulic acid equivalents (ε = 14,700 L m^−1^ mol^−1^, MW = 194.18).

Phlobaphenes were extracted from 10 to 15 mg of freeze-dried *Azolla* fronds by sequentially adding 200 μL of concentrated HCl and 800 μL of dimethyl sulfoxide (DMSO) and applying vigorous vortexing after each of these additions. Extracts were then centrifuged at 14,000 rpm and 4 °C for 45 min, and the supernatants were diluted with methanol (20% final concentration). The concentration of phlobaphenes was expressed as the value of absorbance recorded at their λmax (510 nm) per g of fresh weight, as reported by Landoni et al. [103].

Soluble and insoluble proantocyanidins were extracted from ~10 mg of freeze-dried *A. filiculoides* fronds and quantified, as reported by Li et al. [104].

### 4.5. Chlorophyll Fluorescence Measurements

Chlorophyll fluorescence parameters were measured with an Imaging Pam M-series fluorimeter (Heinz Walz GmbH, Pfullingen, Germany). *Azolla* plants were adapted to dark conditions for 30 min before a saturating light flash (~6000 μmol m^−2^ s^−1^) was applied for less than a second to measure the PSII quantum yield (Fv/Fm), in accordance with Genty et al. [105]. Then, light response curves were constructed by increasing the actinic light to 11, 21, 26, 56, 81, 111, 146, 186, 231, 281, 336, 461, 531, 611, and 700 μmol m^−2^ s^−1^. Under each light intensity, the values of the electron transport rate (ETR), effective PSII quantum yield (ΦPSII), non-photochemical quenching (NPQ), and the coefficient of non-photochemical quenching (qL, which indicates the fraction of open PSII reaction centers) were calculated in accordance with Genty et al. [105], Maxwell and Johnson [106], and Baker [107], as follows:ETR = (Fm′-Ft)/Fm′ × Light intensity × 0.5 × Abs (=0.84)
where Fm′ is the fluorescence maximum with respect to the applied light intensity; Ft is the steady-state level of fluoresce prior to the application of saturation; Abs is the absorptivity, which was set to a standard value typically measured in green leaves (0.84, assuming 84% of the incident photons of photosynthetically active radiation absorbed by the leaf).
ΦPSII = (Fm′ − Ft)/Fm′
where Fm′ and Ft are defined as shown above.
NPQ = (Fm − Fm′)/Fm′
where Fm′ is defined as shown above, and Fm is the dark-adapted value of fluorescence assessed at the plateau level reached during the application of a saturation pulse
qL = (Fm′ − Ft)/(Fm′ − Fo′) × Fo′/Ft
where Fm and Fm′ are defined as shown above, and Fo′ is estimated using the approximation presented by Oxborough and Baker (1997) [108].

All chlorophyll fluorescence parameters were measured simultaneously at 10 different points within the same sample of *Azolla* for each treatment group. Three biological replicates were assayed per treatment group. The mean representative value ± standard error (SE) was then calculated for each experimental condition.

### 4.6. Gene Expression Analysis

#### 4.6.1. RNA Isolation and cDNA Synthesis

RNA was isolated from *A. filiculoides* fronds using the Spectrum Plant Total RNA Kit (Sigma–Aldrich, Milan, Italy) by applying protocol B. Then, it was treated with DNase (Sigma–Aldrich, Milan, Italy) in accordance with the supplier’s instructions. The null PCR amplification in the presence of *Azolla*-specific ITS primers reported by Costarelli and coworkers [25] verified the absence of any DNA contaminating the RNA preparations. In accordance with the instructions of the supplier, 3 µg of RNA was reverse-transcribed in the presence of Maxima H Minus Reverse Transcriptase (Thermo Fisher Scientific, Milan, Italy) and 100 pmol of random hexamers (Euroclone, Milan, Italy).

#### 4.6.2. Identification of Candidate Genes of the Phenylpropanoid Pathway and Quantitative RT-PCR

Regulatory *WDR* and *bHLH* genes of the phenylpropanoid pathway were searched in the *A. filiculoides* protein v1.1. database (https://www.fernbase.org) on 31 March 2021 using the sequences of experimentally validated genes from model and crop plant species as queries in BLAST P. Once retrieved, the sequences of candidate genes were aligned with those of the reference enzymes using MEGA X [109]. The evolutionary history was inferred by using the Maximum Likelihood method and the Whelan and Goldman model [110]. Primer pairs were then designed on selected *WDR* and *bHLH* genes and on *R2R3MYBs* potentially involved in regulating the phenylpropanoid pathway identified by Güngör et al. [43]. For each of the two members of the two *DFR* families present in this fern [44], *DFR1* and *DFR2*, and for the *LAR* gene [43], specific primer pairs were designed. All primer pairs were designed with the help of OligoExpress Software (Applied Biosystems, Waltham, MA, USA). The genes considered in this study and their relative primer pairs are given in Appendix A. An aliquot of 3 µL of 1:10 diluted cDNA was used in the PCR reaction, which was made using the BlasTaq 2X qPCR Mater Mix (ABM, Richmond, Canada) and run in an ABI PRISM 7300 SDS apparatus (Applied Biosystems, Waltham, MA, USA) using the following cycling parameters: an initial step at 95 °C for 3 min, then 40 cycles each including a step at 95 °C for 15 s and a step at 60 °C for 1 min. A melting curve was added after each run. For each gene, four technical replicates were amplified. The efficiency of PCR for each primer pair was tested as reported by Escaray et al. [111], before the application of the (2^−ΔCt^) gene expression quantification method. The ΔCt was based on the differences between the relative expression levels of the target genes and the housekeeping *EF-1α* gene, as characterized by de Vries and colleagues [112], to compare the relative expression profiles among genes and treatments, as reported by Bizzarri et al. [113].

#### 4.6.3. Statistical Analysis

To determine the statistical differences among the treatment groups, six replicates were independently computed. The data were tested for their homogeneity of variance (F-test or Levene’s test) and normality distribution (Shapiro–Wilk’s test). If the assumptions of these tests were not violated, data were analyzed via unpaired two-sample *t*-tests or a one-way analysis of variance (ANOVA) and post-hoc comparison (Tukey’s HSD). If the assumptions of the homogeneity of variance or normality distribution tests were violated, data were analyzed via non-parametric statistics. Statistical analyses were carried out in R studio (version 3.5.3). The level of significance was set to *p* < 0.05 unless stated otherwise, and treatment mean values ± standard errors (SEs) were plotted. Correlation analyses between the expression levels of regulatory and structural genes was performed using the Pearson test.

## Figures and Tables

**Figure 1 ijms-24-08554-f001:**
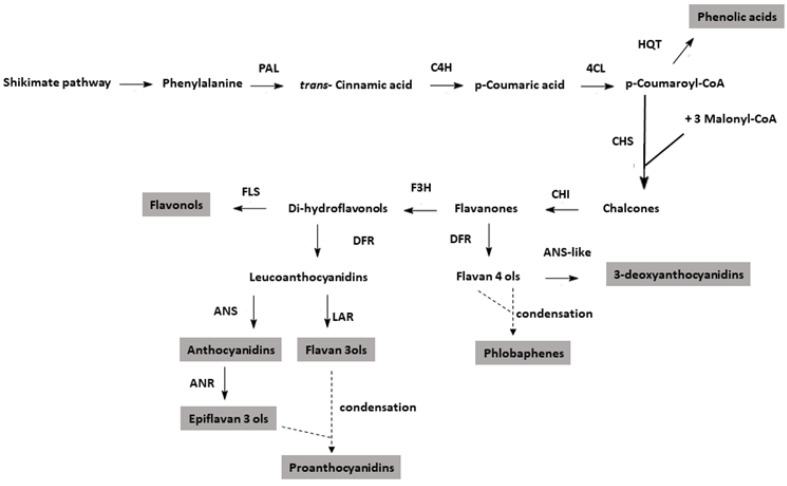
Phenylpropanoid biosynthetic pathway. Only the enzymatic steps significant for the studies presented here are indicated. The gray box presents the compounds assayed in this work.

**Figure 2 ijms-24-08554-f002:**
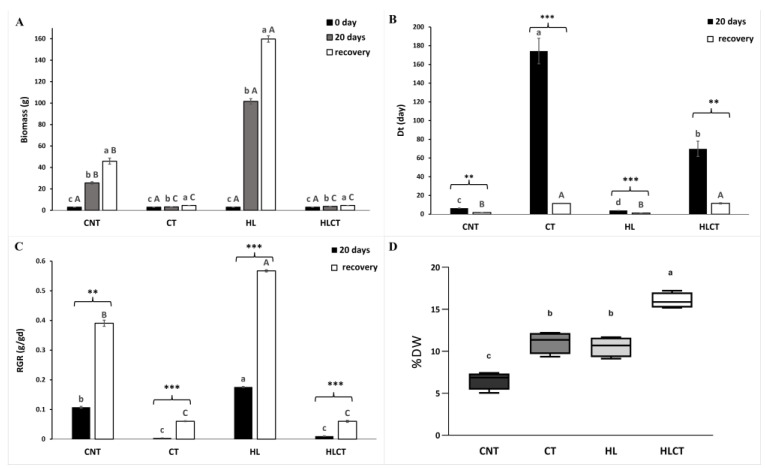
Growth indexes of *A. filiculoides* plants after 20 days of treatment with different light and temperature conditions followed by 7 days of recovery. (**A**) Fresh biomass; (**B**) doubling time (Dt); (**C**) relative growth rate (RGR); (**D**) percentage of dry weight. Data were collected from two independent experiments with three biological replicates each. Significant differences were determined by ANOVA (*p* < 0.01) followed by Tukey’s HSD (*p* < 0.05) or Student’s *t*-test (*p* < 0.01); *n* = 6. In panel (**A**), the statistical significance among the mean values measured at different time points within the same treatment are given with lowercase letters, while the uppercase letters indicate differences among the four treatments at the same time point. In panels (**B**,**C**), statistical significance among the samples after treatments and following recovery are given with lowercase and uppercase letters, respectively. The levels of significance within each treatment are given with asterisks: ** (*p* < 0.01) or *** (*p* <0.001). In panel (**D**) the statistical significance (*p* < 0.01) among the different treatments is given with lower case letters. Treatments are labeled CNT (control); CT (cold treatment); HL (high light intensity); and HLCT (high light intensity and cold treatment).

**Figure 3 ijms-24-08554-f003:**
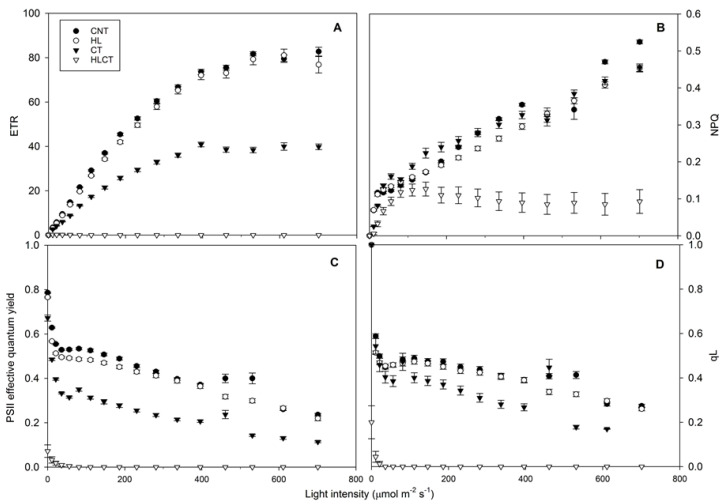
Light response curves of chlorophyll fluorescence parameters. (**A**) Electron transport rate (ETR); (**B**) non-photochemical quenching (NPQ); (**C**) effective PSII quantum yield (ΦPSII); (**D**) coefficient of photochemical quenching (qL). Data are the mean ± SE of data (*n* = 30) collected from two independent experiments with three biological replicates of *A. filiculoides*: under control conditions (CNT, black circles); after exposure to a high light intensity (HL, white circles); after exposure to cold treatment (CT, dark triangles); after exposure to a high light intensity and cold treatment (HLCT, white triangles). The dataset also includes technical replicates.

**Figure 4 ijms-24-08554-f004:**
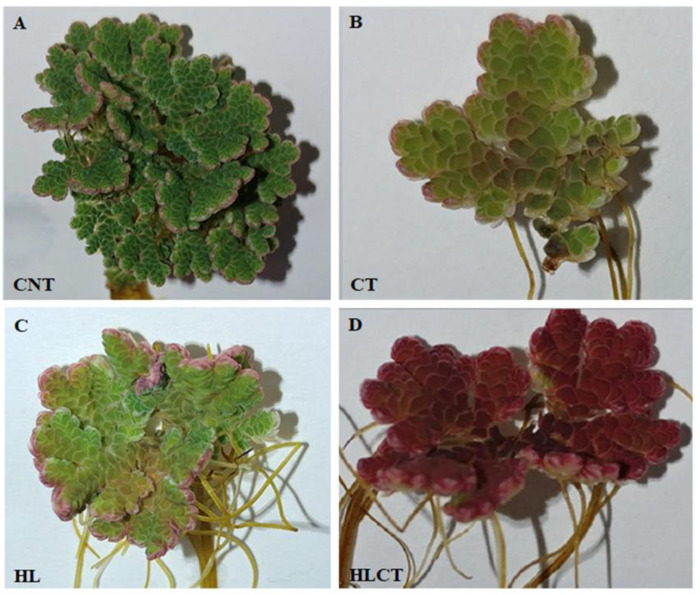
Pigmentation of *A. filiculoides* resulting from different light and temperature treatments. Treatments are labeled as: (**A**) control (CNT); (**B**) cold treatment (CT); (**C**) high light intensity (HL); (**D**) high light intensity and cold treatment (HLCT).

**Figure 5 ijms-24-08554-f005:**
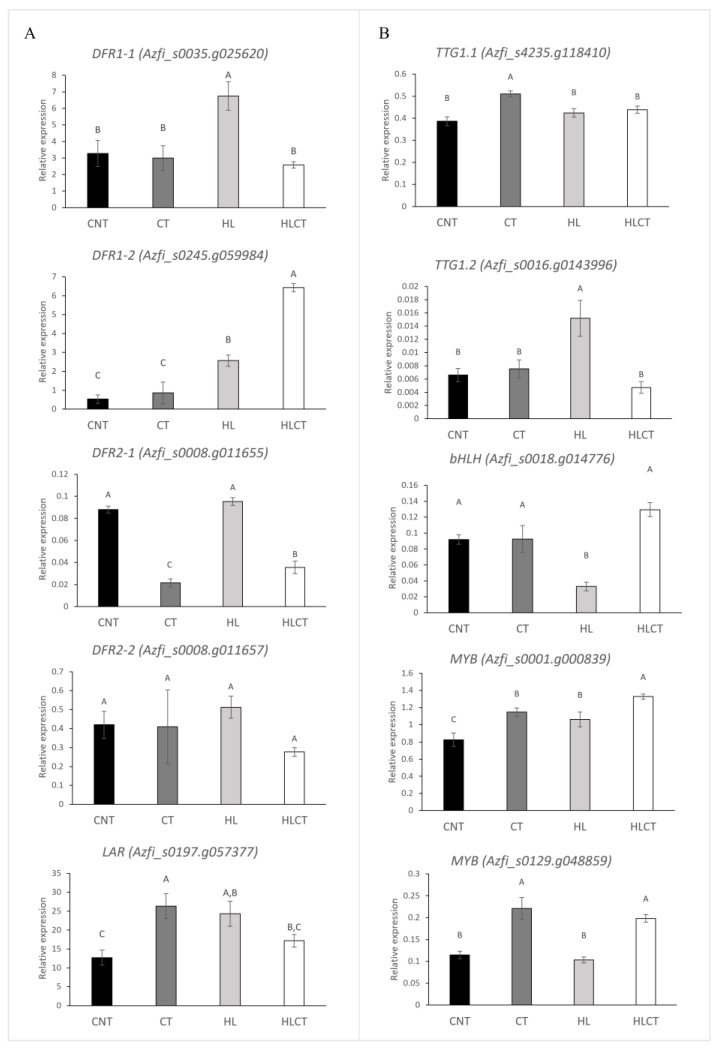
Relative expression levels of selected structural (**A**) and regulatory genes (**B**) in *A. filiculoides* across treatment groups. Data were collected from two experiments with three biological replicates each. The relative expression of each gene was calculated using the (2^−ΔCt^) algorithm. Significant differences determined by ANOVA followed by Tukey’s multiple comparison test (*p* < 0.05; *n* = 6 for both) are indicated by different letters. Treatments are labeled as shown in the legend of Figure 2.

**Table 1 ijms-24-08554-t001:** Measurements of the maximum quantum efficiency of PSII after the dark adaptation (Fv/Fm) of *A. filiculoides* under different conditions of light and temperature. Data were collected from two independent experiments with three biological replicates each. The dataset also includes technical replicates. Statistically significant differences between mean values were assessed by multiple comparison procedures (ANOVA, *p* < 0.001 followed by Tukey’s HSD, *p* < 0.001; *n* = 10). Different letters indicate statistically significant differences among treatments. Treatments are labeled as shown in the legend of Figure 2.

	Fv/Fm	
CNT	0.786 ± 0.009	a
HL	0.765 ± 0.009	a
CT	0.651 ± 0.058	b
HLCT	0.216 ± 0.061	c

**Table 2 ijms-24-08554-t002:** Levels of chlorophylls and carotenoids in *A. filiculoides* under different light and temperature conditions. Data were collected from two independent experiments with three biological replicates each. Significant differences were determined by ANOVA (*p* < 0.001) followed by Tukey’s HSD (*p* < 0.05); *n* = 6. Treatments are labeled as shown in the legend of Figure 2.

	Chl *a* (µg/gFW)		Chl *b* (µg/gFW)		Chl *a*/Chl *b*		Carotenoids (µg/gFW)	
CNT	252.01 ± 15.24	a	45.23 ± 1.40	a	5.55 ± 0.19	ab	100.91 ± 5.80	b
CT	243.14 ± 8.10	a	40.91 ± 1.44	a	5.96 ± 0.18	a	108.53 ± 2.31	b
HL	131.80 ± 7.52	b	27.25 ± 1.28	b	4.85 ± 0.22	bc	96.11 ± 4.06	b
HLCT	145.00 ± 3.25	b	33.467 ± 0.71	c	4.341 ± 0.13	c	148.60 ± 3.35	a

**Table 3 ijms-24-08554-t003:** Phenylpropanoid levels in *A. filiculoides* fronds under different light and temperature conditions assessed through spectrophotometric analyses. Median value ± SE (*n* = 10). Data were collected from two independent experiments with three biological replicates each. The dataset also includes technical replicates. Significant differences determined by ANOVA (*p* < 0.05) followed by the Tukey’s HSD (*p* < 0.05) are indicated by different lowercase letters. Treatments are labeled as shown in the legend of Figure 2. ^1^ Anthocyanidins were quantified as cyanidin 3-glucoside equivalents ^2^ Flavonols were quantified as quercetin 3-glucoside equivalents ^3^ Phenolic acids were quantified as ferulic acid equivalents.

	Anthocyanidins ^1^ (nmol/gDW)	Phlobaphenes (Abs_510_/gDW)	Flavonols ^2^ (µmol/gDW)	Phenolic Acids ^3^ (µmol/gDW)	Apigeninidin (nmol/gDW)	Luteolinidin-5-*O* glucoside (Abs_496_/gDW)	Soluble Proanthocyanidins (g Catechin/gDW)	Insoluble Proanthocyanidins (µmol/gDW)
CNT	755.19 ± 85.58 b	83.76 ± 7.12 bc	9.94 ± 0.62 a	29.46 ± 3.87 a	1396.10 ± 132.83 b	27.56 ± 2.58 b	7.26 ± 0.64 b	16.18 ± 1.51 a
CT	785.30 ± 53.74 b	51.08 ± 6.28 c	7.94 ± 0.67 a	20.66 ± 1.87 ab	1303.84 ± 120.55 b	23.43 ± 2.54 b	13.16 ± 2.16 ab	10.63 ± 0.84 b
HL	1421.57 ± 156.24 a	113.42 ± 5.64 b	11.16 ± 1.66 a	20.44 ± 1.82 ab	3219.77 ± 422.11 a	62.95 ± 6.44 a	17.00 ± 1.54 a	13.76 ± 1.59 ab
HLCT	1629.25 ± 53.15 a	159.33 ± 20.34 a	12.04 ± 1.22 a	15.16 ± 0.83 b	3961.63 ± 139.00 a	76.88 ± 2.22 a	17.04 ± 2.04 a	12.23 ± 1.00 ab

**Table 4 ijms-24-08554-t004:** Correlations between the expression levels of regulatory and structural genes in *A. filiculoides*. Data were collected from two independent experiments with three biological replicates each. Light gray cells indicate significant correlations (*p* < 0.05). Dark gray cells indicate highly significant correlations (*p* < 0.01).

		*DFR1-1*	*DFR1-2*	*DFR2-1*	*DFR2-2*	*LAR*
* **MYBs** *	*Azfi_s0001.g000839*	−0.292	0.701	−0.599	−0.373	0.103
	*Azfi_s0016.g014344*	−0.230	0.057	−0.553	0.131	0.382
	*Azfi_s0129.g048859*	−0.659	0.307	−0.928	−0.458	0.131
* **bHLH** *	*Azfi_s0018.g014776*	−0.823	0.318	−0.517	−0.584	−0.532
* **WDRs** *	*Azfi_s4235.g118410*	−0.177	0.023	−0.700	−0.230	0.552
	*Azfi_s0016.g014399*	0.720	−0.206	0.484	0.501	0.408

## Data Availability

The data presented in this study are available on request from the corresponding author.

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
