# Peer review of "Impact of High Light Intensity and Low Temperature on the Growth and Phenylpropanoid Profile of Azolla filiculoides"

_ijms, 2023, doi:10.3390/ijms24108554_

Round 1

Reviewer 1 Report

Authors investigated the combination effects of high light and low temperature on the growth, photosynthesis, phenylpropanoid and the expression of some candidate genes of phenylpropanoid pathway. The work is a novel study and they showed the significant effect of low temperature and light intensity on growth and flavonoid production of A. filiculoides, respectively. Add some details on used experimental design (CRD or RCBD) and data analysis.

Author Response

Authors investigated the combination effects of high light and low temperature on the growth, photosynthesis, phenylpropanoid and the expression of some candidate genes of phenylpropanoid pathway. The work is a novel study and they showed the significant effect of low temperature and light intensity on growth and flavonoid production of A. filiculoides, respectively.

We thank the reviewer for his/her positive response.

- Add some details on used experimental design (CRD or RCBD) and data analysis.

Thanks for raising this point. Indeed, we used a Completely Randomized Design (CRD), due to both the homogeneity of our experimental units (i.e. Azolla plants) and the limited number of treatments (i.e. higher/lower light intensity and/or higher/lower temperature) and, because of this, we have analysed the data through ANOVA. More details on the experimental design adopted and how data analysis has been performed have been added in the revised version.

Reviewer 2 Report

I kindly invite the authors to check the points mentioned below:

-Title is not appropriate, can be changed to this title" Impact of high light and low temperature on growth and pie nylpropanoid profile of Azolla filiculoides".

- It is important to observe uniformity of writings throughout the text; some places the article is written as Azolla filiculoides, while other places it is written as Azolla, for example lines 17, 18, 20, 26.

-Abstract should be more concise highlighting the main objectives, results obtained and conclusion.

-I suggest reducing the number of keywords to five and removing these: MYB; bHLH; DFR.

-Line 31, 39, 52, 59 …: Azolla spp, " spp " should not be italic.

- The introduction section should include references to recent literature that examines the

 Combination effects of temperature and light on plants.

-At the end of introduction, the aim of the study should be more detailed, and novelty is questionable. Justify novelty in Introduction section.

-The quality of the figures is poor. For examples figure 2.

-Figure 3: Each treatment icon should be described in the figure description.

-Table 1: In table number 1 and figures, it is better to use lowercase letters to indicate significance and non- significance. Please check the whole manuscript and correct them.

-Line 173: For the first time and for all abbreviations, authors should add full name of all provided abbreviations. For example. Chl a and Chl b. Please check the whole manuscript and correct them.

-Table 2: I suggest authors present total chlorophyll.

- Numbers should be rounded to two decimal places, for example tables 1, 2. Please check the whole manuscript and correct them

-Lines 170-172, 181, 282, 210 should be presented below the table. Please check the whole tables and correct them.

- SE amount in the figures and supplementary file should be controlled, especially in figure 5 (LAR).

-Scientific names should be italicized line for example 283: A. filiculoides.

- Temperatures of 5°C or cold stress in day/ night were considered constant?

- What was the reason for considering a temperature of 5 â—¦C and not another between 5 and 25 â—¦C?

-In the material and method section does not clear how the experiment was designed

-Please add the time that the samples were placed at 90 â—¦C.

-Line 553: I would suggest using the chlorophyll a (Chl a) and chlorophyll b (Chl b) abbreviation.

-The statistical analysis section should be separated from section 4.6.2.

-Moreover, I kindly invite the authors to check punctuation and spelling because there are some errors due to a lack of consistency in the manuscript. Please be consistent with the space between number and unit.

-Style of writing of the references is not uniform.

-General revisions of English are suggested.

Author Response

I kindly invite the authors to check the points mentioned below:

- Title is not appropriate, can be changed to this title" Impact of high light and low temperature on growth and phenylpropanoid profile of Azolla filiculoides".

The title has been changed according to the reviewer’s suggestion.

- It is important to observe uniformity of writings throughout the text; some places the article is written as Azolla filiculoides, while other places it is written as Azolla, for example lines 17, 18, 20, 26.

We understand the reviewer’s concern. However, we do not believe the use of a double terms (i.e. Azolla filiculoides, and Azolla) is inconsistent to indicate Azolla plants. Indeed, we respectfully point out that we already made use of such terminology in our previous works (Costarelli et al. 2021; Brilli et al. 2022).

- Abstract should be more concise highlighting the main objectives, results obtained and conclusion.

We understand the reviewer’s concern. However, due to the limited number of words available (only 200), we have made our best to highlight our main objectives (lines 16-24), results achieved (lines 24-29) and conclusions (liens 29-30). Nevertheless, we have a bit reformulated the abstract to meet the reviewer’s request.

- I suggest reducing the number of keywords to five and removing these: MYB; bHLH; DFR.

Following the reviewer’s suggestion, we have removed MYB, bHLH and DFR from the keywords previously listed, and we added “MBW complex”.

- Line 31, 39, 52, 59 …: Azolla spp, " spp " should not be italic.

We thank the reviewer for pointing this out, and we have amended the text as suggested.

- The introduction section should include references to recent literature that examines the combination effects of temperature and light on plants.

To the best of our knowledge, there is no study specifically investigating the impact of both higher/lower temperature and higher/lower light intensity on Azolla’s growth and phenylpropanoid synthesis. Nevertheless, we have already reported in the introduction those studies reporting the effect of either different temperature (reference [8]) or light intensity [references [10, 11, 12,13] on Azolla’s growth, as well as the effects of low temperature and high light intensity in the phenylpropanoid profile of different plant species (references coupled [26,27,28,29]). However, we have now added more references about this topic.

- At the end of introduction, the aim of the study should be more detailed, and novelty is questionable. Justify novelty in Introduction section.

Following the reviewer’s suggestion, we have modified the end of the introduction section.

- The quality of the figures is poor. For examples figure 2.

Following the reviewer’s comment, Figure 2 has been redrawn. Moreover in our submission, we have uploaded, in separate files, all the figures that meet the standards required by the journal.

- Figure 3: Each treatment icon should be described in the figure description.

Following the reviewers’ comment, we need to remark that all the icons indicating different treated Azolla plants have been already indicated in the caption of Figure 3 as: A. filiculoides under control conditions (CNT, black circles), after exposure to high light intensity (HL, white circles), after exposure to low temperatures (CT, dark triangles), after exposure to high light intensity and low temperature (HLCT, white triangles) (lines 164-167).

- Table 1: In table number 1 and figures, it is better to use lowercase letters to indicate significance and non- significance. Please check the whole manuscript and correct them.

Following the reviewers’ comment, we have changed into lowercase letters to indicate statistically significant differences in Table 1 and Figure 5 and make these consistent with other figures/Tables. However, because in Figure 2 are given the results of multiple comparisons, the significances are also given with uppercase letters and asterisks.

- Line 173: For the first time and for all abbreviations, authors should add full name of all provided abbreviations. For example. Chl a and Chl b. Please check the whole manuscript and correct them.

We have amended the text as suggested by the reviewer.

-Table 2: I suggest authors present total chlorophyll.

Thanks for rising this point. However, since we followed the protocol described by Wellburn et al. (1994) for the spectrophotometric analyses of the chlorophylls, in which are given only the equations for the separate calculation of Chl a, Chl b, we did not provide data of the total chlorophylls. On the other hand, we believe it is not correct to simply sum up the values of Chl a and Chl b to obtain the total chlorophylls. Moreover, it is our opinion that providing the value of total chlorophyll does not add new relevant information to our study.

- Numbers should be rounded to two decimal places, for example tables 1, 2. Please check the whole manuscript and correct them

Following the reviewers’ comment, we have rounded the numerical values in Table 2 to make them consistent with those of Table 3. However, we prefer to leave the numerical value of Table 1 with 3 digits, to maintain their accuracy, as always displayed in the literature.

- Lines 170-172, 181, 282, 210 should be presented below the table. Please check the whole tables and correct them.

We have checked the IJMS formatting guidelines and we do confirm that the caption of tables should be presented as a header, whereas captions to figures should be presented as a footer, exactly as we have already done.

- SE amount in the figures and supplementary file should be controlled, especially in figure 5 (LAR).

As requested by the reviewer, we have double-checked the SE values shown in all our figures and supplementary files and we do confirm they are all right. The only mistake we found regards the letter indicating the significance of LAR expression in HLCT treatment given in Figure 5. Indeed, the correct letters should be ‘BC’, rather that only ‘B’, as shown previously. Hence, Figure 5 has been modified accordingly. Thus, although the levels of LAR mRNAs in HLCT remain slightly higher than in CNT, this difference is not more statistically significant. While we apologize for this error, we have reformulated the end of 2.4 section.

- Scientific names should be italicized line for example 283: A. filiculoides.

Following the reviewers’ comment, we have double-checked that the scientific names we mentioned were italicized. However, we must emphasize that, according to IJMS formatting rules, scientific names that appear in the paragraph headings should not be italicized.

- Temperatures of 5°C or cold stress in day/ night were considered constant?

The cold treatment was run into a growth chamber where we could not change the temperature. Thus, the temperature was maintained constant. We have now specified it in the text.

- What was the reason for considering a temperature of 5 â—¦C and not another between 5 and 25 â—¦C?

We selected the temperature of 5 °C moving from our previous study (Costarelli et al. 2021) in which we reported that A. filiculoides collected from a local pond, after experiencing for several months a minimum temperature ranging from 4°C to 13°C, became reddish.

- In the material and method section does not clear how the experiment was designed.

As also requested by another reviewer, we have now specified in the text that we used a Complete Randomized Design (CRD) due to both the homogeneity of our experimental units (i.e. Azolla plants) and the limited number of treatments (i.e. higher/lower light intensity and/or higher/lower temperature).

- Please add the time that the samples were placed at 90 â—¦C.

We have provided the information requested by the reviewer.

- Line 553: I would suggest using the chlorophyll a (Chl a) and chlorophyll b (Chl b) abbreviation.

As suggested by the reviewer, we have used Chl a and Chl b abbreviations to indicate and chlorophyll a and b, respectively.

- The statistical analysis section should be separated from section 4.6.2.

As requested by the reviewer, we have separated the description of the statistical analysis from section 4.6.2, and placed it into the new sub-section 4.6.3.

- Moreover, I kindly invite the authors to check punctuation and spelling because there are some errors due to a lack of consistency in the manuscript. Please be consistent with the space between number and unit.

Following the reviewers’ comment, we have thoroughly checked punctuation and spelling throughout the manuscript.

- Style of writing of the references is not uniform.

Following the reviewers’ comment, we have double-checked the reference style to make it consistent.

- General revisions of English are suggested.

As suggested by the reviewer, we got a colleague of us native English speaking to read the manuscript.

Reviewer 3 Report

The manuscript by Cannavò et al. describes the influence of high light (700 mkmol photons m2 s-1) (HL) and cold treatment (5 ºC at 220 mkmol photons m2 s-1) (CT) on Azolla filiculoides plants.

The manuscript requires some additional experiments, improving of figure presentation, description and discussion of obtained results, in my opinion. Thus, the manuscript in the current view requires major revision.

I listed my main comments below.

The abstract and Introduction are written well. However, I did not understand why the authors abbreviated ‘low temperature’ by CT – L16 (cold treatment-L520). Probably using LT (low temperature) can be more appropriate.

In addition, CNT has no description before use (L111). I realized that it can mean ’control’ but this is extremely unusual to abbreviate this word and seems to be laboratory slang. It should be corrected. The same for Dt (L134), which was not described before use (Figure descriptions does not cover this).

I doubt the right presentation of the statistic on Fig 2A indicated by capital letters, especially in the case of HL (just compare with HLCT).

I want to pay attention here to the data from the 7 days recovery period (Fig 2, white column). This is a correct point of the plan of the experiment, in my opinion. However, the author completely ignored the data, which can be obtained from such plants in the Figs presented below. I think that this is a very significant mistake and these data should be added.

In Fig3 the authors indicated that ETR, effective quantum yields (Y(II) or ФPSII) and probably Y(NPQ) significantly decreased at CT. This is a very usual effect obtained at low temperature and can be explained by lowering of thylakoids membrane fluidity. Especially if take into account that the usual environment for plants is warm and they do not have mechanisms for changing the saturation of the fatty acids of their thylakoids. In addition, if summarized together Y(II), Y(NPQ) at 220 mkmol photons m2 s-1 (CNT)  it will be 0.3+0.3=0.6. I.e. the part of Y(NO), non regulated NPQ, which usually reflectes the PSII photoinhibition, is 1-0.6=0.4. However, the authors wrote about the absence of photoinhibition (L390-393). I think this is incorrect.  

In the case of HLCT the situation is more unclear. If take into account that ETR is equal to 0 and Y(II)=0, then how can the authors talk about the photosynthetic activity of PSII. The main part of PSII is inactivated. The Fv/Fm=0.2 clear indicates it. Moreover, the Y(NO) is near 0.9. The main question here is whether the photoinhibition of PSII is reversible or not? But the data from the recovery period is absent. They are strongly required here. The same is required for Fv/Fm (Table 1) as well as for all below data.

The authors do not indicate (but should) the equations for fluorescence parameters calculations, however this is needed. For example, ETR is calculated as ETR(II)=PAR(II)·Y(II)/(Fv/Fm). Thus, ETR should reduce with Y(II) (or ФPSII) and the sentence in the manuscript ‘the values of ETR were reduced in parallel with those of the PSII efficiency (ΦPSII)’ is not important (L152-153).  

In the discussion, the authors do not take their attention that low temperature can facilitate the ROS production near PSII able to destroy it. It should be added.

Why did the authors extract pigments and centrifuged the obtained solvent in cooled acetone? This is not necessary according to my knowledge, but maybe I do not know something. The authors can see Lichtenthaler1987-ChlorophyllsandCarotenoidsPigmentsofPhotosyntheticBiomembranes or Porra et al., 1989 (BBA, 975,384-394).

Author Response

The manuscript by Cannavò et al. describes the influence of high light (700 mkmol photons m2 s-1) (HL) and cold treatment (5 ºC at 220 mkmol photons m2 s-1) (CT) on Azolla filiculoides plants. The manuscript requires some additional experiments, improving of figure presentation, description and discussion of obtained results, in my opinion. Thus, the manuscript in the current view requires major revision. I listed my main comments below.

- The abstract and Introduction are written well. However, I did not understand why the authors abbreviated ‘low temperature’ by CT – L16 (cold treatment-L520). Probably using LT (low temperature) can be more appropriate.

We thank the reviewer for this suggestion. However, we opted for the use of CT (to mean Cold Treatment) to indicate the treatment at low temperatures instead of LT that could be misleading, as we have already used ‘L’ to abbreviate ‘Light’.

- In addition, CNT has no description before use (L111). I realized that it can mean ’control’ but this is extremely unusual to abbreviate this word and seems to be laboratory slang. It should be corrected. The same for Dt (L134), which was not described before use (Figure descriptions does not cover this).

We decided to indicate the control treatment with ‘CNT’ instead of the more canonical ‘CTRL’ for the same reason as above, to avoid any misunderstanding with ‘L’ that we have already used to abbreviate ‘Light’.

As also requested by another reviewer, we have now specified all the treatment abbreviations when first mentioning them at the beginning of the ‘Results’ section, by using the definition give later on in the M&M section. In addition, following the reviewer request, we have also repeated the meaning of ‘Dt’ in the text, although the definition of ‘Dt’ is already present in the caption to Figure 2 when describing panel ’B: Doubling time (Dt)’.

- I doubt the right presentation of the statistic on Fig 2A indicated by capital letters, especially in the case of HL (just compare with HLCT).

We apologize for not being clear. We have now reported in the legend of Figure 2 a more detailed description of the statistical comparisons to which refer the lowercase and uppercase letters, and the asterisks.

- I want to pay attention here to the data from the 7 days recovery period (Fig 2, white column). This is a correct point of the plan of the experiment, in my opinion. However, the author completely ignored the data, which can be obtained from such plants in the Figs presented below. I think that this is a very significant mistake and these data should be added.

While we understand the point raised by the reviewer, we would like to remark that the ultimate goal of present study was to dissect the impact of higher/lower temperature and/or higher/lower light intensity on growth rate and biosynthesis of phenylpropanoids in Azolla. Therefore, performing a recovery after the light/temperature treatments was instrumental to assess whether the stress(es) we applied were high enough to elicit a severe remodulation of the phenylpropanoid pathway without severely impairing Azolla’s growth. Since during the recovery the doubling time (Dt) and relative growth rate (RGR) values significantly decreased and increased, respectively, with respect to those measured after the CT and HLCT treatments, we can conclude that neither CT nor HLCT treatments irreversibly compromised Azolla’s growth.

- In Fig3 the authors indicated that ETR, effective quantum yields (Y(II) or ФPSII) and probably Y(NPQ) significantly decreased at CT. This is a very usual effect obtained at low temperature and can be explained by lowering of thylakoids membrane fluidity. Especially if take into account that the usual environment for plants is warm and they do not have mechanisms for changing the saturation of the fatty acids of their thylakoids. In addition, if summarized together Y(II), Y(NPQ) at 220 mkmol photons m2 s-1 (CNT)  it will be 0.3+0.3=0.6. I.e. the part of Y(NO), non regulated NPQ, which usually reflectes the PSII photoinhibition, is 1-0.6=0.4. However, the authors wrote about the absence of photoinhibition (L390-393). I think this is incorrect. 

We thank the reviewer for this interesting consideration and we definitely regret not to have run fluorescence measurements after the recovery. However, we fully embrace the reviewer’s thesis that treatment at low temperature and low light intensity might have induced the onset of photoinhibition. However, on the basis of evidence gathered that: a) the value of the maximum quantum efficiency of PSII after dark adaptation (Fv/Fm) slightly decreased (~18 %) with respect control plants; b) the values of doubling time (Dt) and relative growth rate (RGR) measured after recovery were lower and higher, respectively, than those after in CT treatment, we can undoubtedly assess that the effects induced by photoinhibition were transient and did not permanently impaired the growth of Azolla plants. We have reformulated the text to integrate this point in the revised version of the manuscript.

- In the case of HLCT the situation is more unclear. If take into account that ETR is equal to 0 and Y(II)=0, then how can the authors talk about the photosynthetic activity of PSII. The main part of PSII is inactivated. The Fv/Fm=0.2 clear indicates it. Moreover, the Y(NO) is near 0.9. The main question here is whether the photoinhibition of PSII is reversible or not? But the data from the recovery period is absent. They are strongly required here. The same is required for Fv/Fm (Table 1) as well as for all below data.

As we have already discussed in the manuscript, the higher content of polyphenolic compounds induced by high light at low temperature (HLCT) might have interfered with the measurements of chlorophyll fluorescence parameters. On the other hand, due to the slow turnover of phenolic compounds, if we would have run chlorophyll fluoresce measurements after the short period (7 days) of recovery, the results would have likely be affected by this very same inconvenience. Notwithstanding, as reported in the manuscript, the high content of polyphenolic compounds might have avoided irreversible photooxidative damage to the reaction centers of PSII. This hypothesis has been unambiguously confirmed by the biomass and growth parameters (Dt and RGR) measured after recovery which resulted similar in Azolla that experienced low temperature/low light (CT), or low temperature/high light conditions (HLCT). The absence of severe photoinhibition-induced damage has been further confirmed by the evidence that the chlorophyll did not break down in Azolla after HLCT, as Chl a and Chl a/b values resulted similar to those recorded in Azolla plants treated with high light (HL), as the latter being the condition that maximizes Azolla’s growth.

We thank the reviewer for his/her thoughts about reversible vs irreversible photoinhibition of PSII and changed the text to introduce this point. We intend to specifically investigate the effect of photoinhibition on Azolla plants in a future follow-up of the present study, by combining measurements of chlorophyll fluorescence, photosynthetic- and non-photosynthetic pigments with quantification of reactive oxygen species (ROS) (i.e. H2O2) and antioxidant enzymes that would allow to estimate the degree of photoinhibition induced by high/low light intensity at low temperature.

- The authors do not indicate (but should) the equations for fluorescence parameters calculations, however this is needed. For example, ETR is calculated as ETR(II)=PAR(II)·Y(II)/(Fv/Fm). Thus, ETR should reduce with Y(II) (or ФPSII) and the sentence in the manuscript ‘the values of ETR were reduced in parallel with those of the PSII efficiency (ΦPSII)’ is not important (L152-153). 

Following the reviewers’ comment, we have added the equations to calculate the fluorescence parameters which have been programmed in the Imaging Pam M-series fluorimeter (Heinz Walz GmbH) we used for our measurements. Moreover, we have reformulated the point rose up by the reviewer.

- In the discussion, the authors do not take their attention that low temperature can facilitate the ROS production near PSII able to destroy it. It should be added.

We thank the reviewer for his/her suggestion and we have integrated this point in the text.

- Why did the authors extract pigments and centrifuged the obtained solvent in cooled acetone? This is not necessary according to my knowledge, but maybe I do not know something. The authors can see Lichtenthaler1987 - Chlorophylls and Carotenoids Pigmentsof Photosynthetic Biomembranes or Porra et al., 1989 (BBA, 975,384-394).

Indeed, we did not use cold acetone in our extraction. The protocol we employed for the extraction of chlorophylls and carotenoids was that reported in Brilli et al. (2022).

Reviewer 4 Report

Cannavò et al investigated the effect of high light and low temperature on Azolla filiculoides. To do this they measured these effects on growth, photosynthesis and phenylpropanoid amounts. Furthermore, they investigated the genetic regulation of phenylpropanoid biosynthesis under these conditions.

The research was well executed and the results and discussion presented in an organized and comprehensible fashion. The work is also novel and the conclusion that phlobaphenes are up regulated under high light and low temperature at the expense of phenolic acids biosynthesis is very interesting. Also, the clear acknowledgement of the authors to the limitation of their data regarding the genetic regulation of flavonoid biosynthesis is much appreciated.

Overall I think the manuscript is an important contribution to the field and I recommend its acceptance for publication. There are, however, a few minor, mostly linguistic, suggestions to improve the text.

Line 45. CO2 should change to CO2.

Line 62. Please give a definition for ‟frondˮ.

Line 75. Replace ‟coupledˮ with ‟combinedˮ

Line 111. Please give the description of an abbreviation when first used in the text ‘CTM’.

Fig 2. Are the growth parameters for whole plants or detached fronds? Please specify.

Table 2. Please use only one number after the decimal point. All the other numbers after the decimal point is insignificant.

Table 3. ‘Media’ should be ‘Median’

Table 3. Because of the inaccuracy of spectrophotometric quantification of polyphenols, I suggest changing the heading to ‘Relative levels of phenylpropanoids in …..’. Also, please state in the header of table 3 that anthocyanidins were quantified as cyaniding 3-glucoside equivalents, flavonols as quercetin 3-glucoside equivalents and phenolics as ferulic acid equivalents.

Table 4. Dark grey cells indicate highly significant….

Line 382. On the one hand….

Line 399. These genes not only code for….

Line 419. ….is consistent with what was found in higher…..

Line 421.   paves the way for future….

Line 455. Please give the definition for JA.

Where describing the methods used for polyphenol quantification, lines 562 – 589, please specify whether extractions were made from fresh weight or dry weight.

Line 501. …., as well as to better understand the ….

Line 521. You forgot to put in the amount of PPFD.

Author Response

Cannavò et al investigated the effect of high light and low temperature on Azolla filiculoides. To do this they measured these effects on growth, photosynthesis and phenylpropanoid amounts. Furthermore, they investigated the genetic regulation of phenylpropanoid biosynthesis under these conditions.The research was well executed and the results and discussion presented in an organized and comprehensible fashion. The work is also novel and the conclusion that phlobaphenes are up regulated under high light and low temperature at the expense of phenolic acids biosynthesis is very interesting. Also, the clear acknowledgement of the authors to the limitation of their data regarding the genetic regulation of flavonoid biosynthesis is much appreciated.

Overall I think the manuscript is an important contribution to the field and I recommend its acceptance for publication. There are, however, a few minor, mostly linguistic, suggestions to improve the text.

We thank the reviewer for his/her very positive response and we make our best to accommodate all the suggestions made.

- Line 45. CO2 should change to CO2.

We thank the reviewer for pointing it out and amended the text as requested.

- Line 62. Please give a definition for ‟frondˮ.

Following the reviewer’s suggestion, we have added a few words to define fronds.

- Line 75. Replace ‟coupledˮ with ‟combinedˮ

Done as suggested.

- Line 111. Please give the description of an abbreviation when first used in the text ‘CTM’.

As suggested by the reviewer, we have described the treatment abbreviations when first mentioning them at the beginning of the ‘Results’ section, by using the definition give later on in the M&M section.

- Fig 2. Are the growth parameters for whole plants or detached fronds? Please specify.

As requested by the reviewer, we have specified in the caption to Figure 2 that the growth parameters refer to the whole plants.

- Table 2. Please use only one number after the decimal point. All the other numbers after the decimal point is insignificant

As requested by another reviewer, we have rounded to two digits the numerical values of Table 2.

- Table 3. ‘Media’ should be ‘Median’

We thank the reviewer for pointing it out and amended the text as suggested.

- Table 3. Because of the inaccuracy of spectrophotometric quantification of polyphenols, I suggest changing the heading to ‘Relative levels of phenylpropanoids in …..’. Also, please state in the header of table 3 that anthocyanidins were quantified as cyaniding 3-glucoside equivalents, flavonols as quercetin 3-glucoside equivalents and phenolics as ferulic acid equivalents.

We agree with the reviewer about the inaccuracy of spectrophotometric quantification of polyphenols. On the other hand, we believe it would be misleading to add the term “relative” since this would imply that the levels of a given metabolite analysed in treated plants has been estimated/calibrated against the levels of the same metabolite analysed in the control plants. That was not our case. Thus, we decided to change the heading of Table 3 as follows: “Phenylpropanoid levels in A. filiculoides plants under different conditions of light and temperature assessed through spectrophotometric analyses” that we think it is more appropriate.

- Table 4. Dark grey cells indicate highly significant….

Done as suggested.

- Line 382. On the one hand….

Done as suggested.

- Line 399. These genes not only code for….

Done as suggested.

- Line 419. ….is consistent with what was found in higher…..

Done as suggested.

- Line 421.   paves the way for future….

Done as suggested.

- Line 455. Please give the definition for JA.

The definition of jasmonic acid (JA) has been given in the text.

- Where describing the methods used for polyphenol quantification, lines 562 – 589, please specify whether extractions were made from fresh weight or dry weight.

We thank the reviewer for rising this point. Indeed, all the polyphenols data are refereed to dry weight, as stated in Table 3. Conversely, in the Material & method section, we have neither provided this information nor added correct values about the weight employed for each analysis. We apologize for having overlooked it. We have now carefully revised this part and all the information about the weight of the material employed for each extraction protocol has been corrected.   

- Line 501. …., as well as to better understand the ….

Done as suggested.

- Line 521. You forgot to put in the amount of PPFD.

We have added the missing value of light intensity.

Round 2

Reviewer 2 Report

I kindly invite the authors to check the points mentioned below:

-Please change the title to "Impact of high light intensity and low temperature on growth and phenylpropanoid profile of Azolla filiculoides".

-The scientific name should be change to italic.

-Line 117, 442 and … Please change A. filiculoides to italic.

-Line 254 and … Relative growth rate (RGR) should be first including the full name with the abbreviation and then be presented in the text as the abbreviation. Because before this line in the text, it is presented in an abbreviation form.

-Minor revisions of English are suggested.

Author Response

I kindly invite the authors to check the points mentioned below:

-Please change the title to "Impact of high light intensity and low temperature on growth and phenylpropanoid profile of Azolla filiculoides". The scientific name should be change to italic.

We do confirm that the title is already written like that, with ‘Azolla filiculoides’ in Italic style.

-Line 117, 442 and … Please change A. filiculoides to italic.

In both lines A. filiculoides is not in italic style because, according to IJMS formatting rules, scientific names that appear in the paragraph headings should not be italicized.

-Line 254 and … Relative growth rate (RGR) should be first including the full name with the abbreviation and then be presented in the text as the abbreviation. Because before this line in the text, it is presented in an abbreviation form.

Fixed. The definition of Relative Growth Rate (RGR) first appears in the introduction section where it is given with the full name (line 43).

-Minor revisions of English are suggested.

We have, once again, thoroughly checked the English language throughout the manuscript.

Reviewer 3 Report

I am thankful to the authors of the manuscript for their answers; however, some are not enough for my opinion. I will comment it below in detail. In addition, a new version of the manuscript does not contain colored places, which were changed by the authors. On the one hand, this makes difficult for me to see the changes through the text. On the other hand, this is required by the submission rules. Thus, the current version of the manuscript still requires major revision.

Since I can not see the colored places with changed text, I comment only on the response of the authors.

However, we opted for the use of CT (to mean Cold Treatment) to indicate the treatment at low temperatures instead of LT that could be misleading, as we have already used ‘L’ to abbreviate ‘Light’.

Thus, the authors must write ‘cold treatment’ through the entire text. However, already in the title the authors use ‘low temperature’. Such discrepancy should be avoided. In addition, ‘C’ the authors use in ‘CTRL’. This is a very strange explanation.  

 We decided to indicate the control treatment with ‘CNT’ instead of the more canonical ‘CTRL’

Just write ‘control’ to make the understanding of the text easy for readers.

we can undoubtedly assess that the effects induced by photoinhibition were transient and did not permanently impaired the growth of Azolla plants. We have reformulated the text to integrate this point in the revised version of the manuscript.

Just colored those places.

Author Response

- I am thankful to the authors of the manuscript for their answers; however, some are not enough for my opinion. I will comment it below in detail. In addition, a new version of the manuscript does not contain colored places, which were changed by the authors. On the one hand, this makes difficult for me to see the changes through the text. On the other hand, this is required by the submission rules. Thus, the current version of the manuscript still requires major revision. Since I can not see the colored places with changed text, I comment only on the response of the authors.

We are really sorry for this inconvenience. However, we would like to point out that, in accordance with IJMS rules, we have submitted two different files: one (.word) with visible track changes, and another (.pdf) where all the changes to the text have been accepted to facilitate the reading. We regret the reviewer could not access our file with visible track changes to check all the major revision we have already done. However, this inconvenience did not depend on us, rather we believe that some problems might have occurred to the system that allows reviewers to access all the submitted files. We have advised the editor about this inconvenience. As requested, we have attached, once again, the (.word) file visible track changes.

- However, we opted for the use of CT (to mean Cold Treatment) to indicate the treatment at low temperatures instead of LT that could be misleading, as we have already used ‘L’ to abbreviate ‘Light’. Thus, the authors must write ‘cold treatment’ through the entire text. However, already in the title the authors use ‘low temperature’. Such discrepancy should be avoided. In addition, ‘C’ the authors use in ‘CTRL’. This is a very strange explanation. 

In order to accomplish to the reviewer’s request, whenever possible, we have replaced the term “low temperature” with ‘cold treatment’. However, we neither see it as a strict rule to be followed nor the term “low temperature” as misleading. As for the title, we have already changed as suggested by reviewer#2, and we prefer to leave the contrast between ‘high’ light and ‘low’ temperature which, in our option, contributes to catch the readers’ attention.

- We decided to indicate the control treatment with ‘CNT’ instead of the more canonical ‘CTRL’. Just write ‘control’ to make the understanding of the text easy for readers.

We have clearly defined the control treatment as CNT both in the text, figures and tables. We believe this will not create any misunderstanding to the readers. Indeed, none of the other 3 reviewers that who evaluated the manuscript raised any concern about that.

- We can undoubtedly assess that the effects induced by photoinhibition were transient and did not permanently impaired the growth of Azolla plants. We have reformulated the text to integrate this point in the revised version of the manuscript. Just colored those places.

As mentioned above, we have attached the (.word) file visible track changes.

Round 3

Reviewer 3 Report

The manuscript can be accepted in the current view. 

Author Response

We thank the reviewer for his/her positive response.